∂ | **Open Peer Review** | Human Microbiome | Research Article

# A new hypothesis on BV etiology: dichotomous and crisscrossing categorization of complex versus simple on healthy versus BV vaginal microbiomes

Zhanshan (Sam) Ma[1,2]

**ABSTRACT**  It has been estimated that bacterial vaginosis (BV) influences as many as one-third of women, but its etiology remains unclear. A traditionally held view is that dominance by *Lactobacillus* is the hallmark of a healthy vaginal microbiome (VM) and lack of dominance may make women BV-prone. A more recent characterization is that the human VMs can be classified into five major types, four of which possess type-specific dominant species of *Lactobacillus*. The remaining one (type IV) is not dominated by *Lactobacillus* and contains a handful of strictly anaerobic bacteria. Nevertheless, exceptions to the first hypothesis have been noticed from the very beginning, and there is not a definite relationship, suggested yet, between the five VM types and BV status. Here, we propose a novel hypothesis that assumes the existence of four VM types from dichotomous crisscrossing of "complex versus simple (high-diversity or low-dominance versus low-diversity or high-dominance)" on "healthy versus BV" (the four essential dimensions of VMs). We comprehensively test the hypothesis with 7,958 VM samples by demonstrating the statistically significant differences between the four VM types in community diversity, composition, dominance, heterogeneity, and stochasticity, and by identifying unique/enriched species in each VM type. We further verified the categorization (hypothesis) by using six machine learning (ML) algorithms, including deep learning artificial intelligence (AI) technology, to reclassify the randomly mixed VM samples into four respective types and achieved 85%–100% classification precisions. Our hypothesis provides a foundation for further investigating the etiology and automatic diagnosis of BV based on inexpensive amplicon sequencing and AI technology.

**IMPORTANCE**  BV may influence as many as one-third of women, but its etiology remains unclear. A traditional view is that dominance by *Lactobacillus* is the hallmark of a healthy vaginal microbiome and lack of dominance may make women BV-prone. Recent studies show that the human VMs can be classified into five major types, four of which possess type-specific dominant species of *Lactobacillus*. The remaining one (type IV) is not dominated by *Lactobacillus* and contains a handful of strictly anaerobic bacteria. Nevertheless, exceptions to the first hypothesis have been noticed from the very beginning, and there is not a definite relationship, suggested yet, between the five VM types and BV status. Here, we propose and test a novel hypothesis that assumes the existence of four VM types from dichotomous crisscrossing of "complex versus simple (high diversity or low dominance versus low diversity or high dominance)" on "healthy versus BV." Consequently, there are simple BV versus complex BV.

**KEYWORDS**  BV (bacterial vaginosis), human vaginal microbiome (VM), machine learning, artificial intelligence, unique/enriched species (US/ES), specificity diversity framework (SDF), complexity

Address correspondence to Zhanshan (Sam) Ma, ma@vandals.uidaho.edu.

The author declares no conflict of interest.

"BV (bacterial vaginosis) remains a riddle, wrapped in a mystery, and inside an enigma." (1)

"BV is not a single entity, but a syndrome linked to various community types that cause somewhat similar physiological symptoms." (2)

Bacterial vaginosis (BV) is a high-recurrence vaginal disease with an occurrence/recurrence rate of approximately 30% depending on cohorts of women surveyed (3). The disease is associated with vaginal infections and a variety of pregnancy complications, such as pelvic inflammatory disease, premature rupture of membranes, intrauterine growth restriction, intrauterine fetal demise, chorioamnionitis, endometritis, preterm labor and delivery, postpartum infection, ectopic pregnancy, and tubal factor infertility. BV is also an independent risk factor for the acquisition and transmission of sexually transmitted diseases including human immunodeficiency virus (2, 4). In the early 1990s, clinical microbiologists (5) harnessing the artificial culture technology for bacteria had already postulated ecological interpretations, in particular, the community diversity and species dominance to explain BV etiology. Their efforts were prior to the recent metagenomic sequencing revolution, which enables the identification of virtually all microbes both artificially culturable and uncultivable with rather moderate cost. Nonetheless, significant technological advances have not solved the riddle of BV etiology, despite enormous metagenomic big data generated from extensive studies during the last decade or so (4, 6–26). In the present study, big data sets from 14 of these studies are reanalyzed to propose a new hypothesis on BV etiology.

Traditionally, it has been debated that a community lacking of "good" dominant species, especially *Lactobacillus iners*, is BV prone, and the argument has been extended to suggest that high-community diversity (especially high evenness or equivalently low dominance) is BV-prone (4, 5). These hypotheses were supported by many observations that highly dominant bacterial species especially *Lactobacillus iners* are frequently associated with the vagina of many healthy women (2, 4, 9, 12, 13). However, it has been found that non-*Lactobacillus*-dominant vaginal microbiomes (nLDVMs) can occur in healthy women, which is confirmed by a large cohort study involving 1,000+ healthy women and further analyzed with species dominance network (SDN) by Li and Ma (17, 26). They found that the SDN of the nLDVM is predominantly mutualistic as indicated by predominantly positive correlations in the network, while most competitive (negative) correlations originated from the so-termed BVAB (BV-associated anaerobic bacteria) linked to non-BVAB species. It was the latter (non-BVABs) that may have inhibited the former (BVABs). Furthermore, they found that in the network, *Gardnerella* was suppressed by a mutualistic module of 23 genera, and *Lactobacillus* was inhibited by 15 genera. That both *Gardnerella* (a typical BV risk indicator) and *Lactobacillus* (traditionally assumed "good" species) are inhibited by cohorts of non-BVABs are postulated to establish a protective mechanism in women of nLDVM (26). What was slightly surprising is that *Gardnerella* and *Lactobacillus iners* were actually positively correlated, which means that in the nLDVM or *Lactobacillus*-poor vaginal microbiomes, *L. iners* may actually be "traitors" and could be harmful to women's health potentially. Therefore, the existence of dominance by *Lactobacillus* should not be considered as the hallmark of healthy VMs or BV-risk free (26).

In another frontline, researchers have tried to classify the VMs as distinctive types assuming that some of the types would be associated with healthy individuals and others would be associated with BV patients. According to Ravel et al. and Gajer et al. (9, 12, 13), there are five types of human VMs, from type I to type V, with type IV being further classified as two sub-types. It turned out that, while types I, II, III, and V were dominated by *Lactobacillus iners*, *L. crispatus*, *L. gasseri*, or *L. jensenii*, type IV was not dominated by *Lactobacillus*, and instead, contains a handful of strictly anaerobic microbes and most of which belong to the BVAB. Ma and Li further quantified the five types with species specificity (SS) and species diversity (27), the former is a metric proposed by Mariadassou et al. who synthesizes the species prevalence (distribution)

and abundance across potential habitats of the species, and the latter is familiar biodiversity measures in the form of Hill numbers (of which familiar Shannon entropy is a species case) (28). They confirmed that types I, II, III, and V can be identified by the species specificity of indicator species (*Lactobacillus* spp.), which have the highest specificity values. Type IV is characterized by significantly high species diversity than the other four types, but this type lacks an indicator species. The typing studies of the VMs appear to support the classic hypothesis of BV that the BV-prone microbiome possesses significantly higher species diversity than the other four types, which were assumed to be of low BV risks.

Another converging consensus regarding BV is that BV corresponds to certain disturbed non-equilibrium states in terms of the dynamics of complex ecological systems (2, 9, 12, 13). Following this hint, Ma and Ellison proposed a new concept of dominance, which was different from the existing dominance or diversity concept because it is applicable at both species and community scales (29–31). In other words, besides computing a community dominance metric for a community or microbiota, one can also compute a species dominance metric for each species of the microbiota. The importance of the new dominance is obvious from the previously brief introduction to classic hypotheses regarding BV etiology, namely, dominance or lack of dominance by *Lactobacillus* has been a central concept in BV studies (26). As further introduced below, their dominance concept (metrics) will be one of the key quantitative metrics used in this study. As a side note, Ma and Ellison further developed an SDN framework for studying classic diversity-stability relationship (DSR) since dominance and evenness can be considered as both sides of the same coin—high dominance implies low diversity and vice versa (30, 31). The SDN framework was used to investigate the relationships between community dominance (diversity) and community stability (instability), and the instability can be used as a gauge for dysbiosis. With the SDN framework, it was possible to investigate the community state transitions possibly induced by diversity changes. Furthermore, it was found that there were 12 special trios (network motifs) that occur in the microbiomes of BV patients (30, 31), which can be potential *in silico* biomarkers for predicting the community instability or dysbiosis associated with BV.

From the previous brief review, it is clear that the healthy VM (strictly, the VM of healthy women) can be with/without the dominance of *Lactobacillus*. If the dominance is in effect, then the VM is expected to have lower community (species) diversity, and we term it *simple* community. In contrast, when there is a lack of dominance, the community is expected to have higher community (species) diversity, and we term it a complex community. As to the BV microbiome (strictly, the VM of BV patients), such as the type IV, the community diversity is generally high. If we term such high-diversity communities as complex BV (microbiomes), one would wonder if there is simple BV, that is, VM of BV patients, possibly dominated by a single or handful of BVAB. These four types of microbiota or VMs, that is, simple healthy, complex healthy, simple BV, and complex BV, can be considered as dichotomous, crisscrossing of complex versus simple on healthy versus BV characteristics (dimensions), occupying the four regions of a Cartesian coordinate system (Fig. 1). We postulate that the four VM types should cover the full spectrum of VM systems of both healthy and BV individuals.

Figure 1 diagrammed the new four VM-type hypotheses we propose, analyze, and test in this study. Although our hypothesis is obviously inspired by existing theories/hypotheses, especially by the five types of Ravel et al. (12), and it may even be considered as a revision or extension of existing hypotheses, this study is of two distinctive features. First, our hypothesis explicitly maps health status to VM type without ambiguity, and the four types cover the full spectrum of VM types generated from the dichotomous crossings of the two fundamental properties of VM (healthy versus BV, and complex versus simple). Second, we test our hypothesis (especially the distinctive nature) with rigorous statistical tests (models) at the community/metacommunity level (with community diversity, composition, dominance, heterogeneity, stochasticity), as well as at the species level [unique/enriched species (US/ES)]. We further test the hypothesis

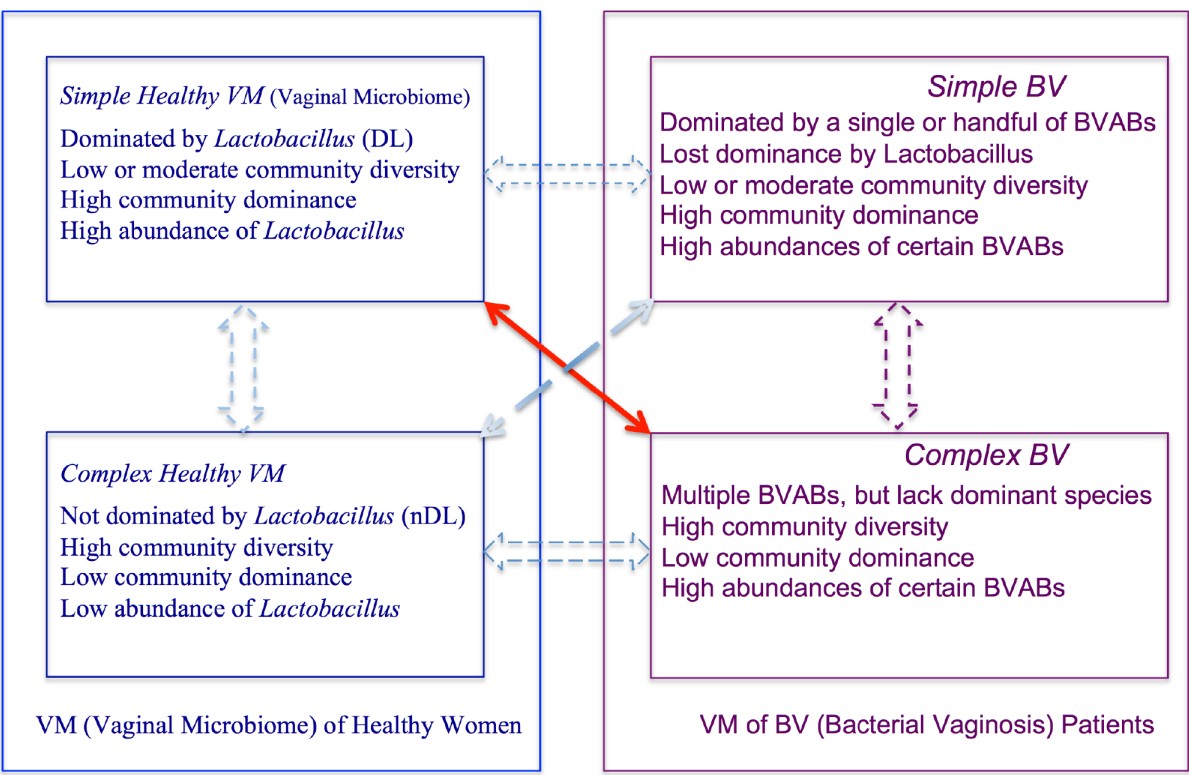

**FIG 1** The hypothesis sketch on BV etiology: dichotomous and crisscrossing of complex versus simple on healthy versus BV vaginal microbiomes. The red arrow signals a traditional view that the loss of dominance by *Lactobacillus* could lead to BV. The *focus* of the proposed hypothesis is on demonstrating the *distinctive nature* (with statistical rigor) of the four VM (vaginal microbiome) types from the said crisscrossing of "complex versus simple" and "healthy versus BV." Note that the focus of this article is **not** on the possible transitions (dashed lines) between the VM types, which is beyond the scope of this article. The terms "complex" (or "simple") were loosely defined as the community with higher diversity or lower dominance (or "lower diversity or higher dominance"), with an implicit assumption that "more" is more complex than "less." See Table 1 for how the four VM types can be quantified and Table 2 for the additional differences between the VM types in composition, heterogeneity, stochasticity, unique/enriched species, nd so on

by reclassifying the randomly mixed samples of all four VM types with six machine learning (ML) algorithms, including powerful deep learning (DL) artificial intelligence (AI) technology. Obviously, the ML algorithms also offer potential clinic technology for the automatic diagnosis of BV.

## MATERIALS AND METHODS

### Data sets of vaginal microbiomes

A total of 14 vaginal microbiome data sets were collected (as shown in Table S1) from published literatures. From the 16s-rRNA raw sequencing reads of the 14 microbiome data sets, we utilized Kraken2 + Bracken to compute the operational taxonomic unit (OTU) tables at the genus and species levels, respectively (32). These two tables were combined into a single OTU table, which includes all species and genera that were uniquely annotated. In addition, if there were species annotated in a genus, and the total abundance of these species (Abundancespecies) was less than the abundance of the genus (Abundancegenus), then the abundance of the genus in the OTU table would be equal to the difference of Abundancespecies and Abundancegenus. Finally, a total of 549 OTUs and 119,843,100 reads were obtained from 3,617 BV samples, and 551 OTUs and 94,641,425 reads were obtained from 3,265 healthy vaginal samples. The BV (3,617 samples) or healthy (3,265 samples) statuses were based on the diagnosis in their original reports. Beyond the above 14 results, we further test the ML models with an

independent data set of 1,076 samples, originally reported by Doyle et al. (17). Hence, a total of 7,958 VM samples are used in this study to test the proposed hypothesis.

We then further classify the healthy group into two sub-classes: healthy complex (HC) versus healthy simple (HS) based on 10 metrics, including the alpha diversity (Hill numbers) from diversity order $q = 0, 1, 2$; community dominance, species dominance of top three species; and species abundance of top three species. Similarly, classify the BV group into two sub-classes: BV complex (BVC) versus BV simple (BVS) based on the same 10 metrics. The sub-classifications are performed using a common clustering algorithm ("*K*-means" function of R software) (R software, Ver. 4.1.0.). The justifications for choosing the 10 metrics, and their computation formulae are introduced in the next sub-section.

## Study design and analysis methods

As illustrated in Fig. 2, the whole study consists of the four steps (i–iv).

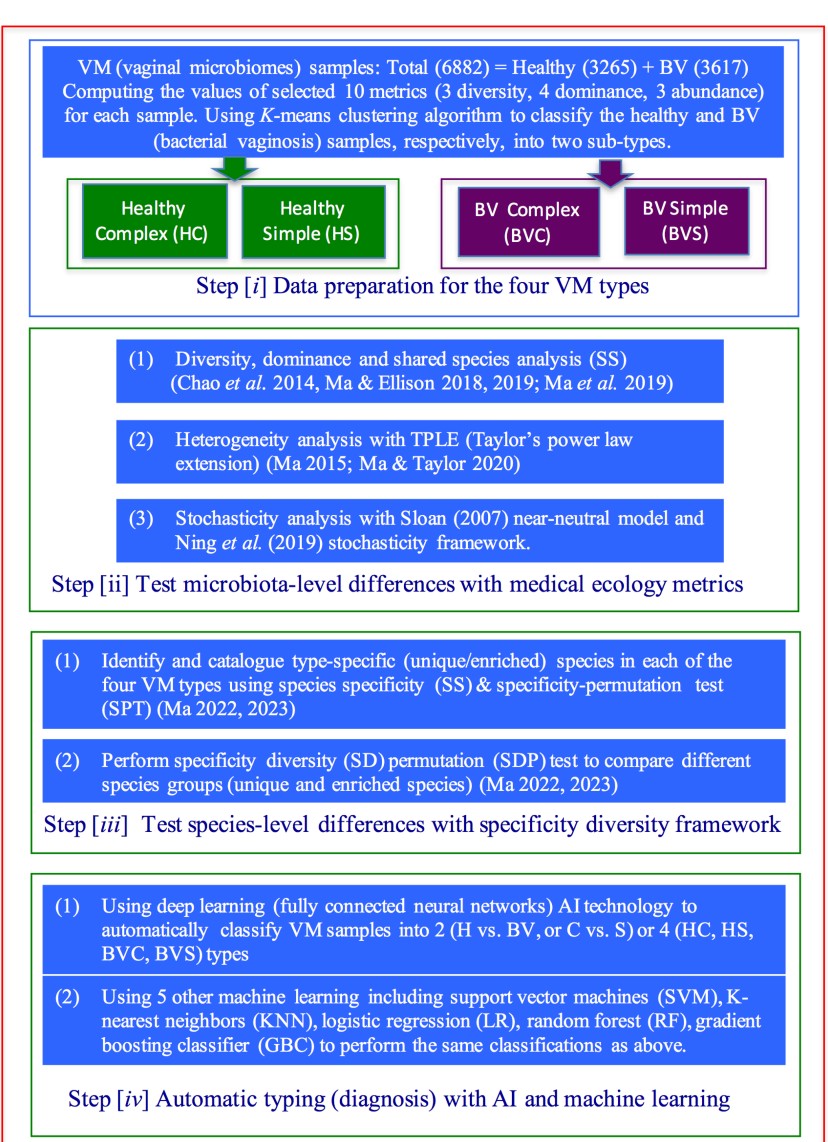

**FIG 2** Study design for formulating, testing, and applying the hypothesis of the four VM types.

### Step i: data preparation for quantifying the hypothesis of the four VM types

Data preparation for sketching out the basic idea (hypothesis) of the four VM types, that is, complex healthy (CH), complex simple (CS), BV healthy (BVH), and BV simple (BVS), required 10 metrics that include three diversity measures (Hill numbers when $q$ = 0, 1, 2), community dominance, species dominance of top three species, and species abundances of top three species. The $K$-means clustering algorithm is used to classify the 3,265 healthy VM samples into HC and HS types (groups) based on the 10 metrics, and, similarly, to classify the 3,617 BV VM samples into BVC and BVS types (groups).

The central idea of the four types of VMs or microbiotas lies in the "superimposition" or "crisscrossing" of a pair of dichotomies, the dichotomy of complex versus simple microbiotas, and the dichotomy of healthy versus BV statuses, as illustrated in both Fig. 1 and 2. The 10 ecological metrics are carefully selected based on our previous experience in analyzing human VMs and in other medical ecology studies (26, 27, 29–31, 33–43). Below is a brief introduction to the 10 metrics and the rationales for their choices.

Community (species) diversity is a *de facto* standard in microbiome analysis since the launch of the human microbiome project, but there are multiple independently developed diversity indexes such as Shannon entropy (evenness) and Simpson index, which may lead to confusion when different indexes present inconsistent trends. We adopt the Hill numbers (44, 45), which is a form of Renyi's entropy and are defined as a series of diversity numbers stratified based on diversity order ($q$ = 0, 1, 2, …) (46). As shown below, Hill numbers at different diversity order ($q$) are weighted differently by species abundance distribution (relative frequency). When $q$ = 0, the diversity (Hill number) defaults to species richness, since the species abundance does not weigh in or ignored. When $q$ = 1, the diversity is an exponential function of Shannon entropy, and the Hill number is weighted in proportion to species relative abundance and represents the equivalent species numbers of common species. When $q$ = 2, the Hill number represents the equivalent species numbers of dominant species, since it is weighted in favor of more abundant species. Hill numbers are defined in the following form (44, 45):

$$qD = \left(\sum_{i=1}^{N} p_i^q\right)^{1/(1-q)} \tag{1}$$

where $p_i$ is the relative species abundance, $q$ is the diversity order, and $N$ is the number of species in the community.

There may be equally good or better community-level metrics than standard species diversity, such as dominance or specificity as explained below, and we believe that community diversity, especially with Hill numbers emphasizing the common ($q$ = 1) and dominant species ($q$ = 2) and rare species ($q$ = 0 or species richness), are still among the best candidates for characterizing the dichotomies (complex versus simple, and healthy versus BV) or the four VM types.

Besides the previous three Hill numbers ($q$ = 0, 1, 2), the fourth metric we adopt is the community dominance concept. Ma and Ellison proposed the dominance concept that is applicable at both community and species levels (29, 30). While dominance and evenness (the main component of diversity) can be considered as both sides of the same coin, the dominance concept (metrics) by Ma and Ellison developed further measures of the "crowdedness" or interactions among species (29, 30). Therefore, in our opinion, dominance can be an important supplement for characterizing the dichotomies of four VM types. The dominance is defined as follows:

$$D_c = m_c^*/m_c = 1 + \sigma_c^2/m_c^2 - 1/m_c \tag{2}$$

where $m_c^*$ is community mean crowding and is defined as:

$$m_c^* = m_c + \sigma_c^2/m_c - 1 \tag{3}$$

where $m_c$ is the mean of the population abundances (size) across all species (i.e., per species) in the community, and $\sigma_c^2$ is corresponding variance. In addition, community mean crowding ($m_c^*$) can also be considered as a measure of community unevenness, and $D_C$ is a linear function of Simpson's diversity index. An additional advantage of $D_C$ is that it can be utilized to define a species dominance metric for each species in the community. To define species dominance metric, Ma and Ellison first defined species dominance distance ($D_{sd}$) in the form of:

$$D_{sd} = m_c^*/m_s = m_c/m_s + \sigma_c^2/(m_c m_s) - 1/m_s \qquad (4)$$

where $m_s$ is the population abundance (size) of the focal species of interest in the community. The species dominance ($D_s$) was defined as the difference between community dominance ($D_s$) and species dominance distance ($D_{sd}$), that is,

$$D_s = D_c - D_{sd}. \qquad (5)$$

$D_s$ is both community- and species-specific, and it measures the relative dominance of a specific (focal) species in the community, and more dominant species possess greater values of $D_s$.

Next, we select species-level metrics for characterizing the dichotomies of VMs, natural selection is the previous species dominance, and we pick the species dominances of the top three most dominant species, that is, the three species with the highest $D_s$ values, or the next three metrics out of the 10 metrics we plan to choose.

Finally, we fill the list of 10 metrics with the relative species abundances of the top three most abundant species.

In summary, we choose 10 metrics, that is, community diversity metrics (Hill numbers at $q = 0, 1, 2$), community dominance ($D_c$), species dominance ($D_s$) of the top three most dominant species, and species abundances of the top three most abundant species. We consider these 10 metrics are of sufficient power to quantify the two pairs of dichotomies (complex versus simple and healthy versus BV) or their full combinations, that is, the four VM types. That is, we believe the 10 metrics are sufficiently powerful to formulate the four-type hypothesis as a testable quantitative model.

The following step (ii) is designed to independently verify the four types classified with the 10 metrics at the community (microbiota) or metacommunity level and further expand the hypothesis with community-level insights. In contrast, step iii is designed to independently verify and expand the hypothesis with species-level characteristics and insights. Specifically, we detect and catalog type-specific US or ES for each of the four VM types. Finally, step iv tests and cross-verifies the hypothesis independently by harnessing the power of AI (DL) and ML methods, that is, automating the typing for diagnosis purposes, a further step toward the practical application of the proposed hypothesis.

### Step ii: testing the holistic (community-level) differences between the four VM types

This holistic verification step employed three groups of approaches/models in community ecology (Fig. 2). Some of the metrics in the first group of approaches in this step, including diversity and dominance, are not truly testing since the metrics tested (diversity and dominance) and are included in the 10 metrics used to quantify the four VM types in step i. While the statistical tests (Wilcoxon test) with diversity and dominance are well expected to distinguish the four VM types with statistical significance (e.g., $P$-value $\leq 0.05$), the shared species analysis is an independent test of the difference in species compositions between the four VM types because species composition is not included in the 10 metrics (34). When the number of shared species is less than the expected number by chance (computed from 1,000 times of random permutations), then it is determined that the community composition is changed between the two VM types.

The second group of approaches in step ii ncludes two approaches, one is Sloan et al.'s near-neutral model, and another is Ning et al.'s stochastic analysis framework (47, 48). Both approaches are designed to determine the relative importance of stochastic drifts (stochasticity) versus deterministic niche selection, with an analogy of the so-termed niche-neutral continuum. With Sloan' near-neutral model, all species within a metacommunity (consisting of source and destination community) can be distinguished into three categories, neutral, below-neutral (negatively selected), and above-neutral (positively selected), and the percentages of each category of species can be computed. Using Fisher's exact test, one can determine, for each species category, whether or not the percentages of neutral, negatively selected, and positively selected species are significantly different between the two VM types. The percentages, ranging between 0% and 100%, reflect the position of a metacommunity on the niche-neutral continuum, with the percentage of positively and negatively selected species representing the deterministic niche selection, and the percentage of neutral species representing the stochastic drifts.

Ning et al.'s stochasticity analysis framework is developed to determine the relative importance of stochasticity versus determinism within a metacommunity by comparing the community similarity with the assumption that the deterministic selection force should drive community more similar or dissimilar than null expectation (48). By repeatedly comparing the observed metacommunity with the null models from sufficiently large number of simulations, one can compute the so-termed normalized stochasticity ratio (NSR) with the value ranging from 0 (totally deterministic without any stochasticity) to 1 (totally stochastic without any determinism). Both Ning's NSR and Sloan's near-neutral model produce the stochasticity level ranging from 0 to 1, but the NSR measures the stochasticity at the community (metacommunity) level, while Sloan's model classifies each species as either neutral, negatively, or positively selected, that is, measuring the stochasticity at the species level (47, 48). Of course, the percentage of each species category in the Sloan model does also gauge the community (metacommunity) level stochasticity.

Obviously, similar to the shared species analysis, both Sloan et al.'s near-neutral model and Ning et al.'s NSR can produce truly independent tests of the differences between the four VMs (47, 48), unlike the community diversity or dominance analysis in the first group of approaches.

The third group approach in step ii, which we apply to test the differences between the four VM types, is Taylor's power law extensions (TPLEs) (36, 49–51). The original TPL (Taylor's power law) was found to describe the spatial distribution of the biological population (49, 50), and the TPLE was found to describe the community spatial heterogeneity (CSH) (36, 51). The heterogeneity and diversity are not the same (52). First, the difference between heterogeneity and diversity can be captured with a motto by Ellison (2020) "zoos are diverse and natural ecosystems are heterogenous," which emphasizes that heterogeneity focuses on interactions between species, while diversity emphasizes the numbers of "discrete" entities (species or species equivalents). Second, heterogeneity is a group property, while diversity can be measured with one entity such as a single community. Without comparing two entities, heterogeneity can hardly make sense.

Taylor discovered that the variance ($V$) and mean ($m$) of population abundances over space follow a power function in the form of (49):

$$V = am^b \qquad (6)$$

where $a$ and $b$ are TPL parameters with $a$ being largely influenced by sampling factors such as the size of a sampling unit, and $b$ being a species-specific property representing the heterogeneity of population spatial distribution. Ma extended TPL from the population to the community level with four TPLEs, from type I to type IV (51). Type-I TPLE measures the CSH, and type II measures the community temporal stability. Type III measures the mixed-species spatial heterogeneity, and type IV measures mixed-species

temporal heterogeneity. TPLEs are of the same mathematical form as the original TPL, but with different interpretations of the model parameters ($a$, $b$) and variables ($V$, $m$). The $m$ in the type I TPLE is the mean species abundance (size) per species in the community, and $V$ is the corresponding variance. In this study, we use type I and type III TPLEs to measure the CSH and mixed-species spatial heterogeneity, respectively, because only spatial heterogeneity is relevant in comparing the four VM types.

### Step iii: type-unique and/or enriched species with specificity diversity framework

The previous step (ii) is designed to test the community (metacommunity) level difference between VM types, and this step (iii) is designed to test the species-level differences by identifying US or ES in each VM type.

Identifying US or ES species in a type is computationally challenging because the problem is similar to the multiple alignment problem in genomics, which belongs to the so-termed non-deterministic polynomial (NP)-hard problems. The challenge lies in that the computational time to obtain the optimum solutions grows exponentially or faster with the increase of the problem size (in the case of this study, the number of species in a VM type). We take advantage of a recent computational advance in specificity diversity framework (SDF) (40), which can efficiently obtain the statistically reliable (with statistical rigor) lists of ES and ES by taking advantage of randomization (permutation) tests guided by entropy (specificity diversity). Specifically, the SDF consists of three components as briefly described below.

The first component of SDF is the species specificity originally proposed by Maria-dassou et al. to characterize the differences among habitats in the abundance-rank distributions (i.e., species abundance distributions) (28). The SS can be considered as a reincarnation of Dufrene and Legendre's indicator index, and the specificity continuum or the values of specificity (ranging between 0 and 1) can be considered as a reincarnation of the classic generalist-species paradigm in microbial ecology of plants and animals (27, 28, 53). Extreme specialists can act as indicator species in the community of a habitat, possessing strong ecological preferences.

Conceptually, species specificity is defined in the context of at least two habitat types; in the case of this study, we have four habitat types or $H = 4$, that is, $h = 1, 2, 3, 4$, corresponding to HC, HS, BVC, BVS, respectively. Each habitat or VM type may have different number of community samples (Table 1), specifically, S(HC) = 1,547, S(HS) = 1,719, S(BV) = 1,627, and S(BVS) = 1,990.

Assume that $M = \begin{bmatrix} a_{ij} \end{bmatrix}$ is the OTU table for the VM samples of a specific VM type, in which $a_{ij}$ is the relative abundance of species $i$ in sample $j$, H = 4 is the number of VM types (or habitats in the original definition of SS). Let $S^h$ be the number of samples from VM type $h$; $S_i^h$ be the number of samples from VM type $h$ where species $i$ is present, $h = 1, 2, 3, H$.

The local species specificity index is defined as

$$\Delta_i^h = A_i^h \times B_i^h \tag{7}$$

**TABLE 1** The average diversity, community dominance, species dominance, and species abundance of four VM groups, a preliminary quantification of the hypothesis (Fig. 1)

| Group | $N$ | Alpha diversity | | | Community dominance | Species dominance | | | Species abundance | | |
|---|---|---|---|---|---|---|---|---|---|---|---|
| | | $q = 0$ | $q = 1$ | $q = 2$ | | Top 1 | Top 2 | Top 3 | Top 1 | Top 2 | Top 3 |
| Healthy-C | 1,546 | 28.534 | 8.098 | 5.548 | 126.898 | 126.264 | 125.007 | 104.679 | 0.354 | 0.229 | 0.120 |
| Healthy-S | 1,719 | 13.440 | 2.314 | 1.901 | 314.748 | 313.865 | 299.055 | 258.606 | 0.646 | 0.286 | 0.036 |
| BV-C | 1,627 | 29.941 | 7.819 | 5.270 | 136.065 | 135.433 | 128.091 | 24.539 | 0.378 | 0.209 | 0.119 |
| BV-S | 1,990 | 14.422 | 2.206 | 1.611 | 364.794 | 363.955 | 339.973 | 303.894 | 0.783 | 0.136 | 0.034 |

where $A_i^h = \frac{S_i^h}{S^h}$, $B_i^h = \frac{<a_i>^h}{\sum_{h=1}^{H}<a_i>^h}$, and $<a_i>^h = \frac{\sum_{j=1}^{S^h} a_{ij}}{S^h}$.

Note that $A_i^h$ is the prevalence of species $i$ in VM type $h$, that is, the fraction of samples from VM type $h$ where species $i$ was found. $<a_i>^h$ denotes the average local abundances of species $i$ in VM type $h$, and $B_i^h$ denotes the abundance share of VM type $h$ in the total population of species $i$.

The specificity $\Delta_i^h \in [0, 1]$ ranges between 0 and 1, a value of 0 indicating that the species is absent in (local) VM type $h$ and a value of 1 indicating that the corresponding species always exists and only exists in that habitat, i.e., a perfect indicator of that VM type. That is, an extreme specialist should satisfy $\Delta_i^h = 1$, which implies it exists in all samples of one VM type only, but nowhere else (in other VM types). The extreme specialist is the perfect indicator species. Extreme generalists should satisfy $\Delta_i^h = 1/H$, which implies that prevalence = 1, with equal abundance in all VM types.

Ma further defined specificity diversity (SD) by applying Renyi entropy to species specificity, with the following formula (40, 46):

$$\,_h^q SD = \left(\sum_{i=1}^{S_h} \lambda_i^q\right)^{1/(1-q)} \tag{8}$$

where $S_h$ is the number of species in habitat or VM type $h$, $\lambda_i = \Delta_i / \sum_{i=1}^{S_h} \Delta_i$ is the relative specificity of species $i$, $q$ is the order number of SD. Obviously, SD is defined for each metacommunity consisting of (some or all) communities in a VM type $h$ (=1, 2, … 4). For each VM type $h$, $\,_h^q SD$ present as a series of SD values, also termed as Hill numbers named after Hill who first introduced Renyi entropy to measure biodiversity, corresponding to different diversity order ($q$) (44). The SD constitutes the second component of Ma's SDF (40).

Besides the previously introduced SS and SD, the SDF depends on the third component, a pair of permutation (randomization) tests that complete the tasks of determining and cataloging type-specific species (US or ES) in each VM type. The specificity permutation (SP) test distinguishes all species of a VM into the following categories of species: US, ES, and non-different species (NS) with statistical rigor, specifically, $P$-value = 0.05 with false discovery rate (FDR) control. The $P$-value is computed based on 1,000 times of random mixing (equivalent to resampling) of all samples from a pair of VMs, for example, HC versus HS. This ensures that the difference in the SS for each species originates from the fundamental difference between the two VM types (e.g., HC versus HS), rather than from random noises. The FDR control further ensures that the permutation test minimizes the false positive (discovery) rate in classifying many species in the metacommunity. In other words, the FDR control further increases the reliability of the US or ES lists.

The other permutation test, specificity diversity permutation (SDP) test, is designed to test the difference between two species categories (e.g., US or ES) in their SD between two VM types. For example, one can test whether or not there is a significant difference in the SD values between a pair of VMs (e.g., HC versus HS) or between a pair of species categories (ES in HC versus ES in HS). Therefore, while the SP test is performed on the species level, the SDP is performed on assembly (community or metacommunity) level.

### Step iv: automating typing (diagnosis) with AI and machine learning

By performing the analyses specified with the approaches (models) outlined in the previous three steps, we aim to demonstrate that the four VM types have significantly different ecological characteristics at both community/metacommunity and species levels. The step (iv) in this section is designed to automatically classify any VM samples into one of the four types correctly. The approaches we use to perform the classification

task come from the popular deep neural network or DL AI and five other ML approaches (54–58).

Different from the traditional deterministic algorithms, ML refers to using computer (machine) instructions (program or software) to learn (improve with experience, usually with more data), which typically consists of three different components: (i) experience (data), (ii) task (output of the algorithm or program, or classification of VM samples in this study), and (iii) objective (performance evaluation of a given output or precision of the classification here) (59).

To design a program capable of learning how to solve a given problem (task), generally two jobs must be done: (i) feature extraction (also known as data representations)—extracting features from training data, which provide input to a model (learning algorithm) and (ii) learning algorithm or program that searches and tries to optimize the model with regarding the objective function from a hypothesis space. In other words, any learning algorithm can be decomposed into three major parts: (i) representation: structuring the hypothesis space, (ii) evaluation of the model (learning algorithm), and (iii) optimization: navigating the hypothesis space.

DL is essentially a renaming, in the new century, of the classic artificial neural network (ANN) that started in the 1960s. The renaming puts it on the frontiers of AI, and many scholars consider it as a field of ML, although the history of ANN may be longer than that of ML. A hallmark of DL or deep neural network is that its representation is similar to the brain network of neurons, which consists of hierarchal layers of neuron network. Much of the ANN or deep neural network have been developed and become established by the 1990s, but it is the big data and hardware acceleration technology (especially the graphic processing unit [GPU] technology) that turn the ANN into arguably the most successful computational intelligence (or soft computing) technology in the new century, surpassing its closest competing paradigms such as evolutionary computation, swarm intelligence, and other ML technology. AI, which was founded by legendary Alan Turing, who was also considered as a founder of modern computer science, from its start, is charged with arguably one of the most ambitious dreams humans have, an intelligent machine that can mimic human intelligence. For this, it should have no surprise that AI has been focusing on different computing paradigms or technologies from reasoning by propositional logic, through expert systems and robotics, computational intelligence, to today's DL that is particularly good at learning from big data.

In metagenomics, previously, ML has been successfully applied to perform OTU clustering, binning, diversity profiling, taxonomic assignment, comparative metagenomics, gene prediction, and functional analysis (59). In the present study, we apply five ML algorithms (approaches) and DL to classify any VM samples into one of the four VM types previously defined, that is, HC, HS, BVC, and BVS.

To address the VM typing problem in this study, we apply six ML algorithms, including DL (specifically, fully connected neural network), logistic regression (LR), $k$-nearest neighbor (KNN), random forest (RF), gradient boosting decision tree (GBT), and support vector machines (SVM). All six learning algorithms use the same inputs, which are generated from the feature extraction (engineering) from the 16S-rRNA sequencing reads, in the form of OTU tables (from bioinformatics pipeline of Kraken2 + Bracken mentioned previously) or the 10 metrics we selected to quantify VM types (as illustrated previously in Table 1). Specifically, we randomly mix the two or four types of VM samples and then split the pooled data sets into two parts: 75% for training (learning) the model and 25% for performance evaluation of the learned model. In addition, we use an independent VM data set of 1,076 samples originally reported by Doyle et al. to further evaluate the results (17). The performance evaluations for all six algorithms are based on the consistency with the four types each sample originally belongs to as determined by the original diagnostics (BV or healthy status) reported in their respective case studies listed in Table S1. The complex versus simple microbiome type is determined by the 10 metrics as specified in Table 1.

A major consideration in selecting the six ML algorithms was the algorithmic sophistication or complexity, from arguably the simplest LR and KNN, through moderate RF, GBT, SVM, to the most sophisticated DL neural network. Furthermore, these methods cover virtually all major types of ML algorithms, including both linear (LR, KNN, and SVM) versus non-linear (RF, GBT, and DL), single stronger predictive model (LR and KNN) versus ensemble of relatively weak predictive models (GBT, RF, and SVM) versus ensemble (network) of strong predictors (DL). In the following, we present an extremely simplified introduction to the LR and ANN (for explaining the concept of DL).

Logistic regression is different from the common regression model given its dependent (outcome) variable $Y$ takes on only one of the two values, for example, "healthy" (0) or "BV" (1), and the probability $P$ of ($Y = 1$) can be represented with the following equation:

$$P\left(Y = 1\right) = 1/\left[1 + e^{-(\beta_0 + \sum \beta_i X_i)}\right] \tag{9}$$

where $X_i$ is the explanatory variable, which can be the matrix of either OTU tables or the 10 metrics displayed in Table 1 and $\beta$ is the parameter vector. The probability of $P$ ($Y = 0$) = $1 - P$ ($Y = 1$)].

The outcome variable can be extended to multiple categories beyond 0 and 1, namely multinomial logistic regression, such as 0, 1, 2, and 3 corresponding to healthy complex, healthy simple, BV complex, and BV simple, respectively. When the probability $P$ can be predicted with the OTU or 10-metric tables $X$, then the model (equation 9) can be used to classify each VM into one of the two or four VM types.

While the previously described LR uses a single predictor model (equation 9), the ANN (as illustrated in Fig. 3) can be considered as a multilayer hierarchical network of the predictor model such as the LR model.

The neural output can be summed with the following equation:

$$f(x_i w_i) = \varphi\left[\sum_i (w_i . x_i)\right] \tag{10}$$

where $x$ and $w$ are the input and weights of the neuron, respectively, and $\varphi$ is the activation function that transforms the weighted inputs into the output of the neuron. In the above neural network, 11 neurons are chained together and form a two-hidden layer, one input and one output layer network. The input layer is fed with OTU tables or 10-metric matrix. A commonly used activation function is the logistic (sigmoid) function, similar to equation 9, with the following equation:

$$\varphi = 1/(1 + e^{-x}) \tag{11}$$

which maps real number $x \in (-\infty, +\infty)$ to the approximate range between 0 and 1.

When the logistic function is adopted as the activation function, the LR is equivalent to a single-input neutral network. A simplified view of an ANN (Fig. 3) can be considered as the network of logistic regression predictors or an ensemble of LR models.

Two intuitive views are rather useful in understanding how a neural network works to learn from the input data and generate the output (Fig. 3). One is that the ANN is rather similar to the chained neurons of our brain, that is, the ANN mimics how our brain works to solve various problems. In other words, the ANN is inspired by how the brain intelligently solves problems, and similarly, the ANN can learn the knowledge from the input data.

Another intuitive view of Fig. 3 is that the ANN can be considered as an abstracted network of a computer program graph, in which the program statements can be considered as the nodes in the ANN, and the control relationships among the nodes can be considered as the connections in the ANN. Similar to how a computer program solves the program by manipulating the information flow, the ANN is essentially a network of

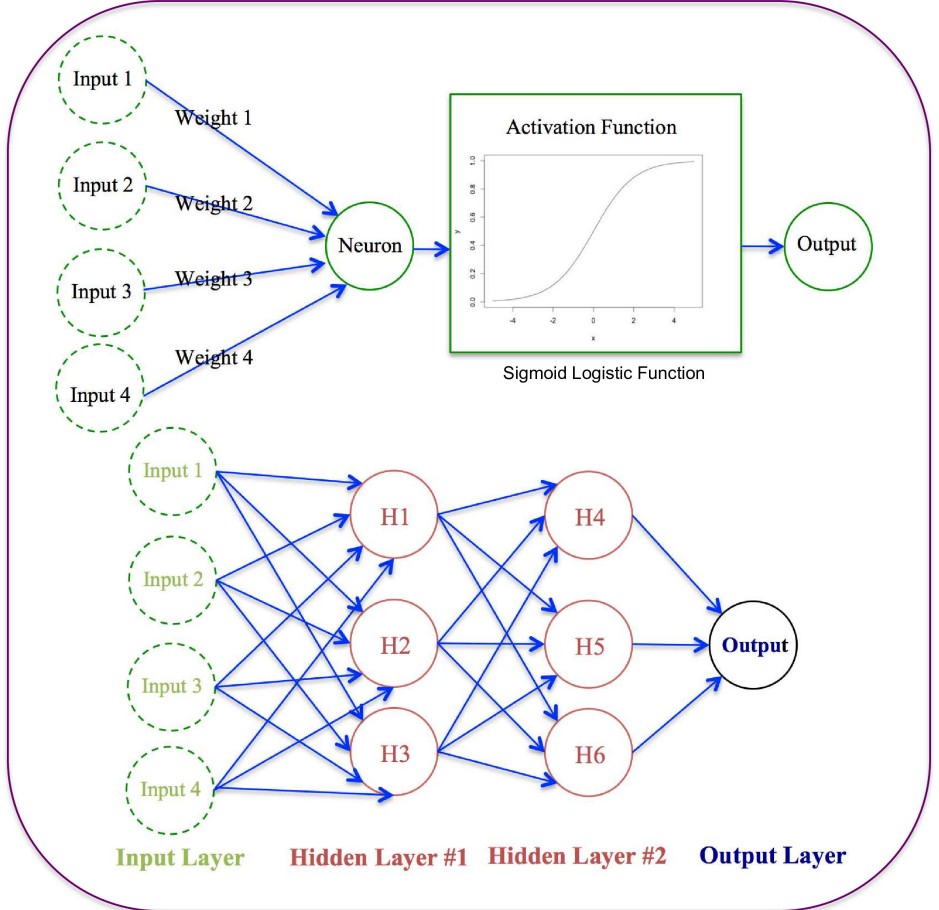

**FIG 3** The abstract structure of the artificial neural network (ANN): the top is the structure of a single artificial neuron (similar to the logistic regression model), and the bottom is a fully connected, multilayer neural network consisting of input, hidden, and output layers.

chained neurons for information processing. Therefore, the ANN and the DL in large can be considered general-purpose information processing machines, and they are essentially computational machines implemented in software programs.

## RESULTS

### Test the four-type hypothesis with holistic microbiota-level metrics/models

#### Diversity, dominance, and composition (shared species) analyses

Using $K$-means clustering algorithm, we classified the 3,265 healthy vaginal samples into 1,546 healthy complex (HC) and 1,719 healthy simple (HS), and 3,617 BV samples into 1,627 BV complex (BVC) and 1,990 BV simple (BVS) samples, respectively. Technically, we used the "$k$-means" function of R software (Version 3.6.3), which adopted the Hartigan-Wong algorithm, to classify the four clusters below (60). Table 1 below exhibits the clustering results and the averages of the 10 metrics used to conduct the classifications. On the surface, the HC and BVC are rather similar to each other, and so do the HS and BVS. This seems to suggest a fundamental characteristic of the human VMs, complex or high diversity community is not necessarily BV-prone, and simple or high dominance (low diversity) community is not necessarily healthy, indicating the necessity of crisscrossing the dichotomies of complex versus simple, and healthy versus BV.

As shown in Table S2 and verified with the Wilcoxon test ($P$-value $\leq 0.05$) that complex communities (HC and BVC) have higher diversity and lower dominance than simple

communities (HS and BVS), regardless of the disease status. This is the main message from diversity and dominance analysis.

However, other comparisons in Table S2 showed intricacies of the diversity patterns. Although there is not a simple pattern, the relationships are consistent with the hypothesis outlined in Fig. 1. For example, comparing HC versus BVC, the former has lower dominance, lower species richness ($q = 0$), but more dominant species ($q = 2, 3$) than the latter in terms of the Hill numbers. There is no significant difference in the typical species ($q = 1$) between HC and BVC.

For another example, comparing HS and BVS, the former again also has lower dominance and lower species richness ($q = 0$), but higher diversity at all other orders ($q = 1, 2, 3$). This reinforces the primary message of the hypothesis—crossing the combination of the complex versus healthy and healthy versus BV is necessary for understanding BV etiology and proper classifications of VMs.

As shown in Table S3, the shared species analysis shows that the four VM types (HC, HS, BVC, and BVS) have distinct species compositions, as indicated by both reads randomization and samples randomization algorithms (34). That is, not only complex and simple VMs have different species compositions but also BV and healthy VMs differ in their compositions, including the crossing combinations of the two dichotomies. Although the shared species analysis indicates the differences, the algorithms are not able to reveal lists of US or ES in each of the four VM types. The following SDF fills the gap, as demonstrated below (40, 61).

## Stochasticity analysis with near-neutral model and stochasticity analysis framework

Table S4A shows the results from fitting Sloan et al.'s near-neutral model (47), and the first two columns shows the metacommunity (consisting of the source and destination communities, between which species migration could occur), and the last three columns show the percentages of below-neutral (negatively selected), neutral (null selection), and above-neutral (positively selected) species in each metacommunity. The average percentages of positively selected (above-neutral) species versus neutral and below-neutral species across all metacommunities are approximately the same 52% versus 48%. This suggests that both deterministic niche selection and stochastic neutrality (drifts) are in effect in human VM, as a previous study indicated (33).

Further analysis with Fisher's exact test suggests more interesting results (Table S4B). First, the percentages of three categories of species (neutral, below, and above-neutral) are significantly different between complex and simple communities, as shown by the first two lines in Table S4B. That is, complexity does affect the balance between stochastic drifts and deterministic selection. Second, disease status does not seem to matter, as shown by the last two columns in Table S4B. Overall, this should be expected since community complexity should be the product of long-term coevolution between microbiomes and hosts, while disease (BV) occurs on a shorter ecological time scale. The evolution should have more far-reaching implications than ecological events such as disease occurrence.

Table S5A shows the results from applying Ning et al.'s stochasticity analysis framework (48): the first column lists the four VM types, the second column lists the number of comparisons made to compute the community similarity (the third column), and the last column lists normalized stochasticity ratio (NSR). The average NSR across four types is equal to 0.365, which is exactly the same as the 36.5% (0.365) of the average percentage of neutral species as revealed by Sloan et al.'s near-neutral model (Table S4A) (47). Therefore, both Sloan's model and NSR cross-verified each other's results. Further analysis with the Wilcoxon test (Table S5B), however, suggests that the stochasticity levels are in general significantly different between the four VM types. This suggests that both complexity and disease status matter in influencing the balance between deterministic selection and stochastic drifts. A natural question is how to reconcile the apparent inconsistency between Sloan's near-neutral model and NSR finding. Our

explanation is that the scale in both methods is different: the neutrality in Sloan's near-neutral model is determined at a species level, and the stochasticity (neutrality) in NSR is determined at a community (metacommunity) level. The difference between both methods was explained in the previous Materials and Methods section.

### Heterogeneity analysis with Taylor's power law extensions

Heterogeneity is different from diversity as explained previously, that is, species interactions are taken into consideration with heterogeneity, while diversity, especially the Hill numbers, measures the effective number of species but their interactions are ignored. Table S6A shows TPLE parameters, and Table S6B shows the $P$-value from permutation tests of the parameter differences between the four VM types. Two findings are particularly interesting. First, the most important parameter, the heterogeneity scaling parameter ($b$) of TPLE exceeds 2 for simple microbiomes, while below 2 for complex communities, regardless of the disease status. This suggests that complexity seems to have more far-reaching implications than disease status, which is consistent with the previous near-neutral modeling. Second, the scaling parameter ($b$) is different between the four types (Table S6B), suggesting that heterogeneity is influenced by both complexity and disease status, which is consistent with the finding from previous NSR results.

### Identify type-unique and enriched species with specificity diversity paradigm

In the previous section, we tested the four-type hypothesis (Fig. 1) by presenting the evidence that the four VM types are holistically different in diversity, dominance, shared species (composition), heterogeneity, and stochasticity at community/metacommunity scale. Here, we further test the hypothesis by presenting the evidence that the four types own their type-specific US and/or ES lists.

Table S7A through F exhibited the species specificity (SS) lists of the four VM types, in pairwise comparisons (six sheers) of the four types. Table S8A through F cataloged the lists of US and ES in each of the four VM types, again in pairwised comparisons. Figure 4 shows the volcano graphs of the five species categories. In each volcano graph, five species categories in each pair of VM types, that is, US and ES in the former (e.g., healthy complex), US and ES in the latter (e.g., healthy simple), and NS (no difference) species (e.g., between the healthy complex and healthy simple). Since there are a total of six pairs of comparisons between the four VM types, corresponding to the six volcano graphs, and all of which show that each VM type has its own US and ES (Fig. 4).

Figure 5 shows the bar-chart graphs of the specificity diversity (SD) ($y$-axis) at each diversity order ($q = 0–4$) of the six species categories (two US categories, two ES categories, fifth category of all significantly different species in specificity, and sixth category of all species without considering specificity), in each pairwise comparison of the VM types. Similar to the previous specificity permutation (SP) tests, from which the US/ES lists are generated, the specificity diversity (SD) permutation (SDP) tests were performed for six pairs of VM comparisons. Figure 5 selected two graphs (Fig. 5A and B) from the six pairs, and the selected two are healthy complex versus BV complex, and healthy simple versus BV simple.

Table S9 further exhibited the results of the SDP tests from the pairwise SD comparisons of the four VM types mentioned previously. As expected, the most significant differences occurred in the two US categories (Table S9A and B), followed by the two ES categories (Table S9C and D), all species with significant differences in SS (Table S9E), and all species (Table S9F) with possibly decreasing significance level. The reason for this potential decrease is obvious, as determined by the nature of the six species categories, from unique species with the largest difference in species specificity to "all species" without considering specificity at all.

Except for the specificity diversity order $q = 0$, and a handful of SD when $q \geq 2$, virtually all comparisons based on the SDP tests revealed a 100% difference between the four VM types. The exceptions are expected since $q \geq 2$ corresponds to the SDs of

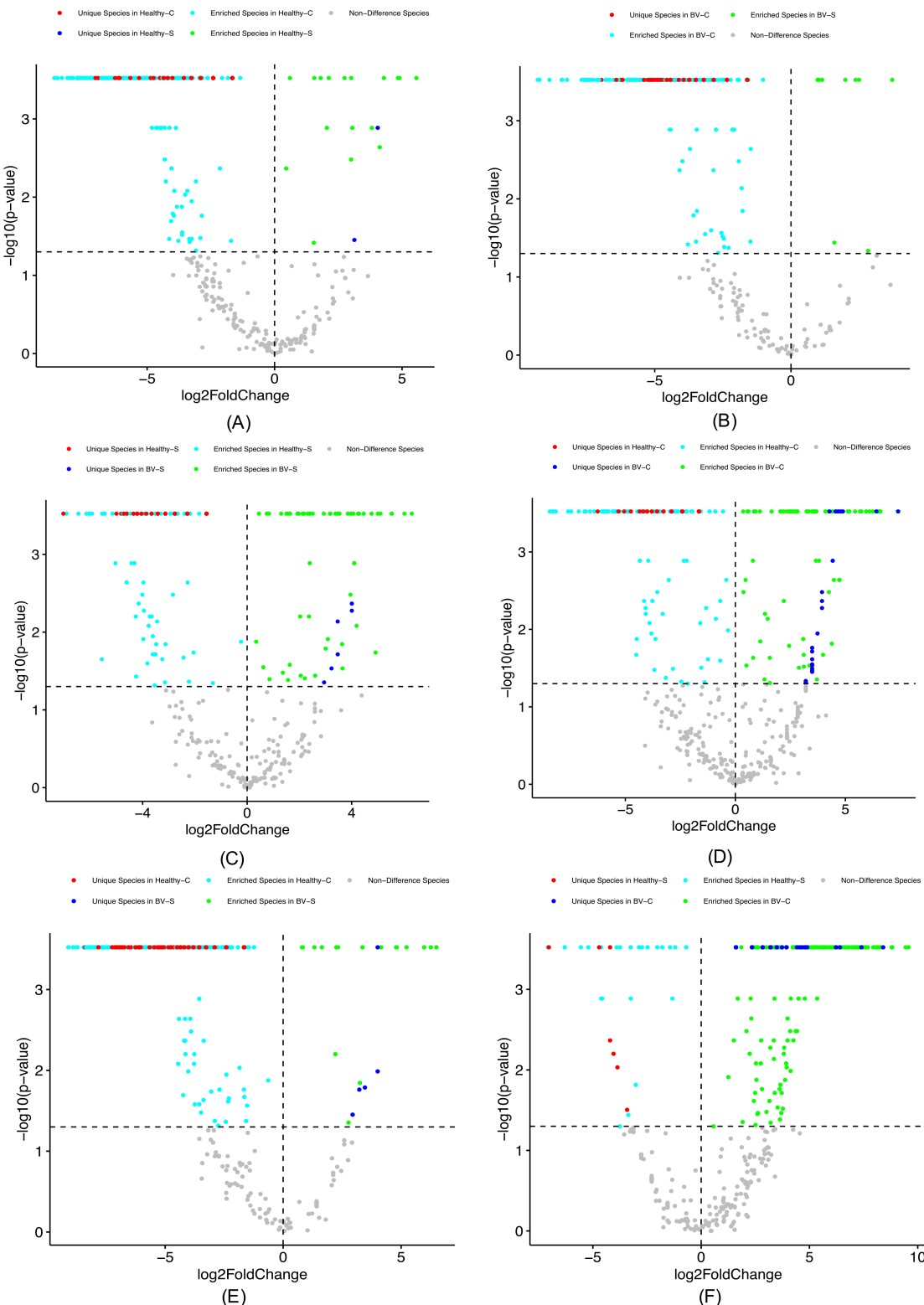

**FIG 4** The volcano graphs of the six pairwise comparisons between the four VM types show the unique/enriched/non-different species in each VM type. The x-axis represents the log-transformation of the specificity fold change between VM types within each pair, where fold change = $S_1/S_2$ ($S$ represents species specificity of type 1 or type 2 or the former type 1 the latter type); the y-axis represents the negative log-transformation of the P-value from SP (specificity permutation) tests of the specificity differences between two VM types. The vertical dotted line at $x = 0$ represents fold change = 1 (i.e., $S_1 = S_2$), the points on the right side of this dotted line represent species with $S_1/S_2 > 1$ (i.e., $S_1 > S_2$), and the left points represent species with $S_1/S_2 < 1$ (i.e., $S_1 < S_2$). The horizontal

**FIG 4** (Continued)

dotted line represents *P*-value = 0.05 [−log$_{10}$(0.05) = 1.301], the points above the line represent species specificity with significant differences between VM type 1 and VM type 2, and the points below represent species of non-significant differences in specificity. Therefore, gray points represent species of non-significant differences in species specificity between the two VM types, cyan points represent significantly enriched species in the former type (type 1), green points represent significantly enriched species in the latter type (type 2), and red and blue points represent unique species in the former and latter types, respectively.

dominant or more dominant species, and both healthy-simple and BV-simple VM types share a significant level of similarity due to the existence of dominant species, which leads to relatively low diversity and is the reason why such communities are considered as "simple." As to the SD at *q* = 0, it is, in fact, the species richness or the number of the species in the species assembly (group or category) for which the SD is estimated since species specificity does not weigh in the computation of SD. Therefore, the comparisons of SD (*q* = 0) between VM types are no different from the comparisons of traditional species diversity as Ma et al., (34) which suggested that the differences in traditional species diversity measured in Hill numbers (including species richness, i.e., *q* = 0) differs only in approximately one-third of the comparisons between the healthy and diseased treatments.

## Automatic typing (diagnosis) of VM samples with AI and machine learning

What is aimed to achieve in this sub-section is somewhat opposite to those reported in the previous two sub-sections, where we demonstrate that the four VM types are of significantly different characteristics as measured by several metrics/models at either community/metacommunity or species levels. In the present sub-section, we demonstrate, with six ML algorithms including DL AI technology, that all VM samples can be automatically classified into one of the four VM types, as defined in our new hypothesis (Fig. 1), with satisfactory precisions.

Here, we first present a brief summary of the running parameters of the six ML algorithms used for the automatic typing of VM samples, that is, for automating the diagnosis. For the DL approach, our network is a simple neural network with an input layer, two fully connected hidden layers, and an output layer, implemented with TensorFlow framework. The learning rate was not a constant, and the initial value was 0.01. We adjusted the learning rate with the parameter "patience" = 30, which means that if 30 iterations could not improve the model performance, the learning rate would be reduced by 10 times (one-tenth of the previous rate). Besides the neural network model, we also performed five other ML models by using the "sklearn" package of Python (Version 3.6). Specifically, KNN requires the number of neighbors (*N*), and we set it to *N* = 17. The parameter of SVM was set as penalty = l2. RF, GBT, and LR did not need hyperparameters when running with the sklearn.

With each of the six ML algorithms, we build (train) seven models, corresponding to the following seven classification tasks: (i) healthy-C versus healthy-S, (ii) BV-C versus BV-S, (iii) healthy-C versus BV-C, (iv) healthy-S versus BV-S, (v) healthy-C versus BV-S, (vi) healthy-S versus BV-C, and (vii) H-C versus H-S versus BV-C versus BV-S. The first six models are designed (trained or learned) to reclassify the pairwise remixture of two VM types, and the seventh is designed to reclassify the remixture of all four VM types. To learn each model, we use 75% of the samples for training the model and the remaining 25% of the samples for testing the model's performance. To avoid the potential bias from arbitrarily splitting the whole data set, we repeat the "75%/25%" splitting of samples 100 times and repeat the model training and testing for 100 times accordingly. A total of (6 methods) × (7 classification tasks) × (100 repetitions) 4,200 ML models were trained and tested. Table S10 tabulated the performance evaluations (precisions) of the 4,200 models in performing the seven classification tasks, that is, automatically reclassifying randomly mixed samples of VM types.

We used two different kinds of input data sets for ML programs: one is the OTU tables and another is the 10 ecological metrics previously selected for defining the four VM types (Table 1). The former input data should be more informative given it contains more

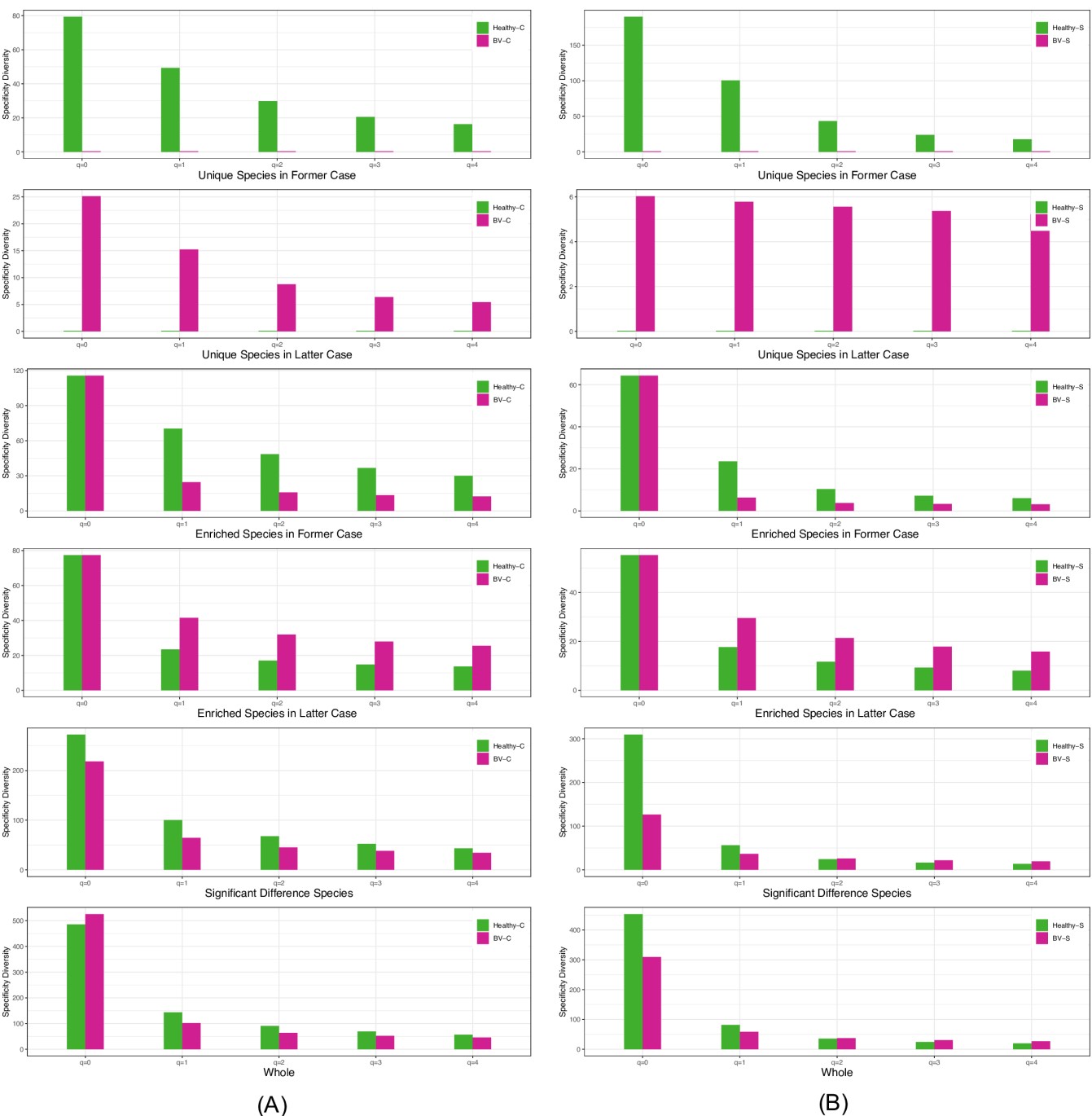

**FIG 5** Specificity diversity (SD) (y-axis) at different diversity order (x-axis, q= 0–4) of the six corresponding species categories of pairwise VM types: (A) the left for the comparison of the six species categories between healthy-C and BV-C (pair) of VM types and (B) the right for the comparisons of the six species categories between healthy-S and BV-S (pair) of VM types. The six species categories include two unique species (US) groups, two enriched species (ES), species with significant differences in SD, and "whole" (all species).

detailed information but may require more computational resources. The latter input data use synthesized metrics summarized from the former data type, and the learning process should be light-weighted.

As shown in Table S10 (also *see* Fig 6), with the OTU tables as input, in terms of the mean precision from 100 times of repetitions, RF (with precision = 0.914), GBT (0.894), and neural network (NN, 0.881) performed slightly better than the other three methods

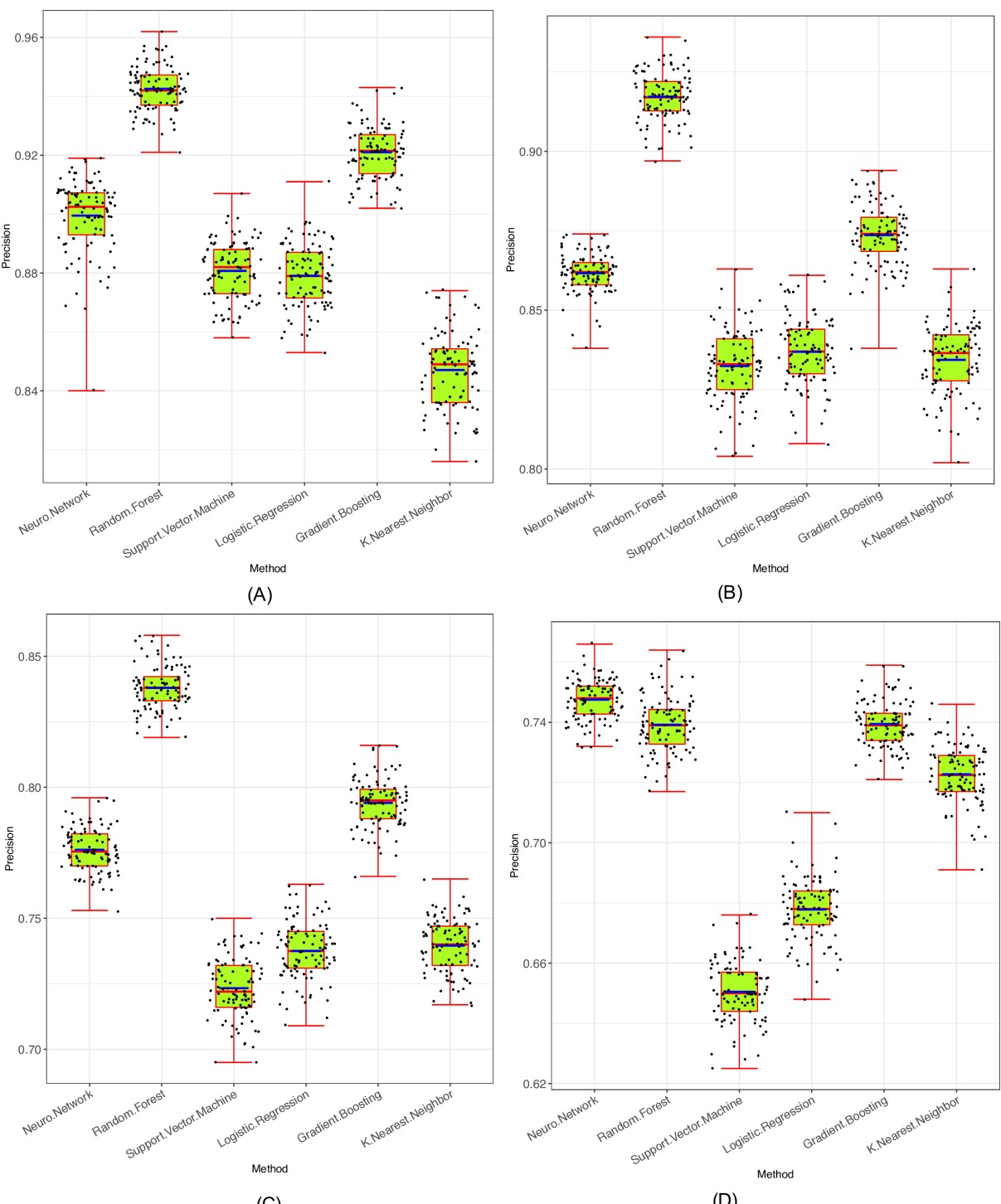

**FIG 6** Performance (precision) evaluations of the six machine learning algorithms in their automatic classifications of the four VM types: (A) healthy-complex versus BV-complex classified with OTU tables; (B) healthy-simple versus BV-simple with OTU tables; (C) four VM type classifications with OTU tables; and (D) four VM types with 10-metrics.

including LR (0.858), KNN (0.847), and SVM (0.850). All six methods demonstrated the precision level of 85% or above. The three better-performed methods (RF, GB, and NN) belong to the so-termed ensemble learning (optimization), which means that they use an ensemble (population or group) of predictive models (such as a number of neurons)

**TABLE 2** Summary on the hypothesis tests of the "four-type VM dichotomies" of complex versus simple and healthy versus BV (all tested differences are of statistical significance with $P = 0.05$)

| VM type | Diversity in Hill numbers (Table 1; Table S2) | Composition with shared species analysis (Table S3) | Dominance (community or species) (Table 1; Table S2) | Heterogeneity with Taylor's and power law extensions (Table S6) | Stochasticity with NSR and Sloan's near-neutral model (Tables S4 and S5) | Unique/enriched species (Tables S7 and S8) | Specificity diversity (SD) (Table S9) | Machine learning with OTU or 10-metrics as inputs (Table S10) |
|---|---|---|---|---|---|---|---|---|
| Healthy-complex | High | Community composition changed significantly = all pairwise shared species numbers lowered | Low | 1. Scaling parameter ($b$) is different between four types. 2. Simple VM types with $b > 2$, and complex VM with $b < 2$. | Complexity does influence the balance between stochastic drifts and deterministic selection, but BV may not. | Table S8 shows the catalogs of unique/enriched species in each VM type. | Virtually all species groups (unique/enriched) and all VM types differ in SD. | With 6 ML algorithms, precisions of classifying four VM types exceed 85%. |
| Healthy-simple | Low | | High | | | | | |
| BV-complex | High | | Low | | | | | |
| BV-simple | Low | | High | | | | | |

to optimize the solutions. The other three methods use non-ensemble strategy (i.e., using a single strong predictive model) and are relatively simple, while the ensemble-based approaches are more sophisticated.

With the alternative 10-metrics input data, the average precisions are almost the same as the average with the OTU tables. However, the range or fluctuations of the precisions in classifying seven different mixtures of VM samples (e.g., healthy-C versus BV-S) are slightly larger than using the OTU tables. For the mixtures of healthy-C versus BV-S reclassification, the precision is nearly 100%. The precisions of the reclassifications of mixed samples from all four VM types are slightly poor than those using the OTU tables, which is due to the fact that the 10 metrics are summarized from the OUT tables and there may be information loss from the summarization (Fig. 6).

Beyond the above training/test results, we further test the ML models with an independent data set of 1,076 samples, originally reported by Doyle et al. (17). The test results are supplied in the bottom section of Table S10. The data set of Doyle includes 825 healthy-complex type and 251 healthy-simple type samples (17). Using the 10-metrics as input data, the ML models trained with previously independent data sets (Table S1; Table 1), and the precisions tested with independent Doyle et al.'s data ranged between 84% and 85%, which are virtually the same as the minimum precision levels obtained from the random splitting of same data sets as training/testing subsets based on "75%/25%" scheme (17). The differences between the two kinds of tests did not exceed 5%.

Finally, we split the very same data set of Doyle et al. with 50:50% of samples as training and testing data set, respectively (17). With this training/testing scheme, the precisions from using 10-metrics reached 98%–100% across all six ML algorithms, but the precisions from using the OTU tables are lower except for RF and GBT algorithms, which did reach the precision level of 85% (Fig 6).

## DISCUSSION

In conclusion, we have demonstrated that the hypothesis of four-type VM dichotomies (complex versus simple, healthy versus BV) is valid through a series of medical ecology and ML algorithms. Table 2 below summarizes the test results from eight aspects starting from diversity/composition/dominance/heterogeneity/stochasticity/specificity analyses, through US/ES lists, to ML algorithms. In particular, the six ML algorithms (especially NN, RF, and GBT) can effectively reclassify the randomly mixed samples of the four VM types into one of the four types, with the precision level exceeding 85%, which has the potential for clinical diagnosis with more extensive training data accumulated in the future.

In perspective, there are three aspects in which we hope to expand and consequently deepen our understanding of BV etiology. First, we hope to further conduct comparative mechanistic studies of complex versus simple VM types. In the literature on theoretical ecology, there is an extensive discussion on the complexity of ecological communities (ecosystems), but almost all of the studies involve stability and time-series data. Although we have previously explored the diversity (dominance) stability relationship with species dominance metrics (29–31), linking the VM typing with stability is not an easy task, given the current scarcity of longitudinal studies (time-series data) on the human VM. The concept (metrics) of dominance is used in this study (four of the 10 metrics used), but much of the SDN analysis, which may link complexity with connectedness and stability, are not explored in this study. In this venue, classic theoretical ecology can play an important role in revealing community complexity and stability (or instability = dysbiosis of VMs) (62). Second, we hope to further improve the ML algorithms with the accumulation of more metagenomic sequencing data of BV-associated microbiomes and develop it as a clinically feasible tool for the automatic diagnosis of BV in practice. In the second venue, the applications of ML can be beyond VM classification and/or BV diagnosis. Specifically, instead of treating ML as a black box, we can use ML and AI as a translucent box for interpreting the underlying mechanisms that may be responsible for

observed patterns (54, 55, 57, 58), for example, the VM types underlying the BV etiology in the case of this study. Third, from the evolutionary ecology perspective, we postulate that complex versus simple dichotomy in the human VM is likely primarily shaped by coevolution between hosts and microbiomes, while the dichotomy of healthy versus disease statuses is more likely driven by events on an ecological time scale (daily basis or at most individual lifespan). Of course, as great ecologist, G. E. Hutchinson had suggested in his classic monograph "The Ecological Theater and the Evolutionary Play" (63), evolution and ecology should be intertwined and it may be impractical for us to separate the two dichotomies in our four VM type hypothesis. Many of the approaches used in this study such as neutral theory and TPL involve both ecological and evolutionary scales (36, 47, 48), but it was indeed difficult to partition both dimensions. Nevertheless, comparative studies with VMs of primates should still be worthy regarding this topic.

Finally, we would like to propose a more ambitious suggestion, beyond this study, regarding the relationship between human microbiomes and diseases. Ever since the launch of the human microbiome projects, diversity analysis has been a *de facto* standard operation routinely computed in virtually all studies on human microbiome–associated diseases. Nevertheless, a comprehensive study by Ma et al. found that when rigorous statistical tests are applied, the reported diversity-disease relationships (DDRs) were significant in only approximately one-third of the cases, which suggest that the DDR is quite uncertain (34). The four-type VM hypothesis tested in this study actually negates a simple monotonic relationship between BV and microbial diversity. A follow-up question can be: is the new hypothesis of general applicability beyond BV? Here, we offer a cautious answer—it depends on the disease system, and it is likely to be applicable to some diseases with similar mechanisms as BV, such as mastitis and dental diseases in our speculation (64, 65). We hope that the methodology demonstrated in this study will be useful for investigating those additional diseases, in which ecological complexity seems to play a "double-faced" but certainly important role.

## ACKNOWLEDGMENTS

I appreciate the computational support from Dr. Lianwei Li and Mr. Xu Yang of Computational Biology and Medical Ecology Lab, Chinese Academy of Sciences. I am indebted to Dr. Larry Forney from the University of Idaho and Dr. Aaron Ellison from Harvard University. It was Larry who inspired and mentored me to study the human microbiome and BV. It was Aaron who encouraged and inspired me to study the medical ecology of human microbiomes. I am also grateful to two anonymous expert reviewers for their insightful comments and suggestions.

This study received funding from the following sources: two National Natural Science Foundation (NSFC) grants (grant no. 31970116 and grant no. 72274192).

The author declares no conflict of interests.

## AUTHOR AFFILIATIONS

[1]Computational Biology and Medical Ecology Lab, State Key Lab of Genetic Resources and Evolution, Kunming Institute of Zoology, Chinese Academy of Sciences, Kunming, China
[2]Center for Excellence in Animal Evolution and Genetics, Chinese Academy of Sciences, Kunming, China

## AUTHOR ORCIDs

Zhanshan (Sam) Ma http://orcid.org/0000-0002-4397-2630

## AUTHOR CONTRIBUTIONS

Zhanshan (Sam) Ma, Conceptualization, Data curation, Formal analysis, Funding acquisition, Investigation, Methodology, Software, Visualization, Writing – original draft, Writing – review and editing

## DATA AVAILABILITY

All data sets reanalyzed in this study can be downloaded from NCBI: https://www.ncbi.nlm.nih.gov/, and a brief introduction to the data sets is listed in Table S1.

## ETHICS APPROVAL

Ethics approval was not necessary since the study does not involve any wet-lab experiments or survey on human or animal subjects and all analyzed data sets are already available in public domain.

## ADDITIONAL FILES

The following material is available online.

### Supplemental Material

**Additional Supplemental Tables (mSystems00049-23-s0001.xls).** Tables S7 and S8.
**Supplemental Tables (mSystems00049-23-s0002.pdf).** Tables S1-S6, S9, and S10.

### Open Peer Review

**PEER REVIEW HISTORY (review-history.pdf).** An accounting of the reviewer comments and feedback.

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
