## [Reviewer comments · mSystems]

A new hypothesis on BV etiology: dichotomous and crisscrossing categorization of complex vs. simple upon healthy vs. BV vaginal microbiomes

Zhanshan (Sam) Ma

Corresponding Author(s): Zhanshan (Sam) Ma, Kunming Institute of Zoology Chinese Academy of Sciences

Review Timeline:

Submission Date:	January 15, 2023
Editorial Decision:	March 24, 2023
Revision Received:	April 28, 2023
Accepted:	June 14, 2023

Editor: Nicholas Chia

Reviewer(s): Disclosure of reviewer identity is with reference to reviewer comments included in decision letter(s). The following individuals involved in review of your submission have agreed to reveal their identity: Ashok Choudhary (Reviewer #1); Douglas J Creedon (Reviewer #2)

Transaction Report:

DOI: <https://doi.org/10.1128/msystems.00049-23>

March 24, 2023

Prof. Zhanshan (Sam) Ma
Kunming Institute of Zoology Chinese Academy of Sciences
Kunming
Kunming
China

Re: mSystems00049-23 (A new hypothesis on BV etiology: dichotomous and crisscrossing categorization of complex vs. simple upon healthy vs. BV vaginal microbiomes)

Dear Prof. Zhanshan (Sam) Ma:

Thank you for submitting your manuscript to mSystems. We have completed our review and I am pleased to inform you that, in principle, we expect to accept it for publication in mSystems. However, acceptance will not be final until you have adequately addressed the reviewer comments.

Preparing Revision Guidelines

Sincerely,

Nicholas Chia

Editor, mSystems

Journals Department
American Society for Microbiology

Reviewer comments:

Reviewer #1 (Comments for the Author):

The hypothesis proposed in this manuscript introduces a novel method to study vaginal microbiomes which is of great importance since bacterial vaginosis impacts a large population. Therefore it is an important step toward gaining a better understanding of the vaginal microbiome environment and its implication for bacterial vaginosis. The authors also make an important conclusion that more complex ecosystems are not necessarily prone to BV.

Although it is an important work toward gaining a better understanding of BV, the author needs to make a few more clarifications regarding the experimental design.

1) The authors have applied the k-mean algorithm to identify 4 clusters. It should be mentioned if this was the best choice for the number of clusters in the data and if different values of k were tried and this was found to be the best.

2) While doing k-mean what was the metric for distance chosen? Did the data clearly indicate just four clusters or the choice of k was based on the number of classes in the hypothesis?

3) The features were generated using 10 metrics so was there any normalization performed on the features before doing k-mean? Did the authors explore and see if the number of features required to make the classification changed with the choice of a number of features?

4) Did the authors try any dimensionality reduction and then do the k-mean to obtain the four classes?

Once the four classes are obtained the author does further analysis. The evaluated quantities show some unique characteristics for each class proposed in the hypothesis. Additional test further validates the existence of four classes. The results and conclusions drawn are consistent with the data provided in the supplementary material.

The application of supervised machine learning models to identify each VM class seems to perform well and looks consistent with the expectations. More details about the choice of hyperparameter are required.

1) When using a neural network what kind of architecture was chosen? What was the values learning rate? Did the author vary the architecture and examined the performance?

2) More details about the hyperparameter in various supervised machine learning algorithm is required.

3) Have the authors tried performing k-mean with the OTU input data? If yes, do they produce the same four clusters?

Overall the authors need to provide clarification on the above-mentioned choices.

Reviewer #2 (Comments for the Author):

In this manuscript the author proposes a novel framework for the healthy versus BV vaginal microbiome. Instead of restricting the healthy states to a particular lactobacillus dominance, the author proposes a framework of diversity as a measure of complexity of the VM in both healthy and BV states. The author uses existing VM databases to identify the four proposed states of healthy-simple, healthy-complex, BV-simple, and BV-complex. This hypothesis represents a next step where it considers not simply a dominant species to categorize the VM, but begins to consider the complex interactions among organisms in both the healthy and BV states.

Specific comments:

1) Please discuss the decision to limit to 2 states of diversity (simple vs. complex) as clearly the VM is not as "simple" as once thought.

2) When we consider vaginal vaginitis - it is clear that the patient symptoms and laboratory findings are an end state that can be attributed to specific pathogenic organisms. Please address in the discussion the difference in thinking of vaginitis versus vaginosis and why a species specific approach to BV should not be pursued. I agree that it is not likely monotonic, but it is still possible that it is multiple different VM states that constitute BV - this would make a "unifying" hypothesis a no/go from the start.

Dear Dr. Chia and Expert Reviewers,

Thank you very much for your time and efforts on reviewing my manuscript! I appreciate your comments and suggestions, which have been meticulously incorporated into the revised version.

Included below please find my point-by-point responses to your review notes. Please noted that my responses were in different fonts (Segoe UI) from your original decision letter (Time New Fonts).

Please feel free to inform me if you have any additional advice for improving my work. I look forward to receiving your favorable decision in due course.

Sincerely Yours,

Sam Ma

PhDs (Computer Sci. & Entomology)
Professor and Director
Computational Biology and Medical Ecology Lab
Chinese Academy of Sciences

=====

March 24, 2023

Prof. Zhanshan (Sam) Ma
Kunming Institute of Zoology Chinese Academy of Sciences
Kunming
China

Re: mSystems00049-23 (A new hypothesis on BV etiology: dichotomous and crisscrossing categorization of complex vs. simple upon healthy vs. BV vaginal microbiomes)

Dear Prof. Zhanshan (Sam) Ma:

Thank you for submitting your manuscript to mSystems. We have completed our review and I am pleased to inform you that, in principle, we expect to accept it for publication in mSystems. However, acceptance will not be final until you have adequately addressed the reviewer comments.

Preparing Revision Guidelines

To submit your modified manuscript, log onto the eJP submission site at <https://msystems.msubmit.net/cgi-bin/main.plex>. Go to Author Tasks and click the appropriate

manuscript title to begin the revision process. The information that you entered when you first submitted the paper will be displayed. Please update the information as necessary. Here are a few examples of required updates that authors must address:

Sincerely,

Nicholas Chia

Editor, mSystems

Journals Department
Reviewer comments:

Reviewer #1 (Comments for the Author):

The hypothesis proposed in this manuscript introduces a novel method to study vaginal microbiomes which is of great importance since bacterial vaginosis impacts a large population. Therefore it is an important step toward gaining a better understanding of the vaginal

microbiome environment and its implication for bacterial vaginosis. The authors also make an important conclusion that more complex ecosystems are not necessarily prone to BV. Although it is an important work toward gaining a better understanding of BV, the author needs to make a few more clarifications regarding the experimental design.

Response:

Thank you very much for reviewing my manuscript! and I particularly appreciate your insightful comments and revision suggestions, for which I present my specific answers below. Please feel free to let me know if you have any further advice on my responses below.

1) The authors have applied the k-mean algorithm to identify 4 clusters. It should be mentioned if this was the best choice for the number of clusters in the data and if different values of k were tried and this was found to be the best.

2) While doing k-mean what was the metric for distance chosen? Did the data clearly indicate just four clusters or the choice of k was based on the number of classes in the hypothesis?

Response:

Please allow me to answer both questions on one place.

Indeed, K-means algorithm may not be the best approach to determine the number of clusters. The approach was adopted to formulate the initial hypothesis and the primary consideration was convenience and its simplicity. Regardless of the number of clusters it may produce, the results will be either confirmed or rejected by the comprehensive tests (including other 6 machine learning algorithms) in later stages. Some of the 6 machine learning algorithms are supervised learning and some are unsupervised learning. If all tests confirmed the four-cluster results, then the initial choice of K in K -means becomes a moot issue.

It was actually the case that all consequent tests confirmed the four-cluster classification. For this, I simply specify the four clusters in K-means clustering to formulate the initial hypothesis and prepare the datasets for further tests in later stages. The major finding of the paper is then that all consequent tests confirmed the hypothesis. So, I did not do additional tweaking with the initial K-means clustering.

Technically, I used the " K -means" function of R-software (Version 3.6.3) to classify the four clusters. The algorithm " K -means" function used is "Hartigan-Wong" (Hartigan and Wong, 1979). Hartigan, J. A. and Wong, M. A. (1979). Algorithm AS 136: A K-means clustering algorithm. *Applied Statistics*, 28, 100-108. doi: 10.2307/2346830

3) The features were generated using 10 metrics so was there any normalization performed on the features before doing k-mean? Did the authors explore and see if the number of features required to make the classification changed with the choice of a number of features?

Response:

The 10 metrics were carefully selected based on our previous studies on diversity, heterogeneity, stability, ... basically cover all major aspects of community ecology we could conceive with the strongest relevance in our judgement. However, the number 10 per se was somewhat artificial, for example, the top 3 dominant species, could be top 2 or top 4, then the number of metrics would be different. Thank you very much for raising this interesting issue.

The input to K-means were raw metrics values, and no specific normalization was adopted. However, the K-means algorithm internally uses PCA-kind algorithm to normalize the metrics so that all dimensions (metrics) can be clustered on same scale (unit).

4) Did the authors try any dimensionality reduction and then do the k-mean to obtain the four classes?

Response:

Not really, as mentioned previously, but the PCA-like nature of the K-mean algorithm also has certain dimension-reduction effects. The choice of ten metrics was based on ecological insights, and the dimension of 10 was intentionally controlled to be moderate. The apparently loose choices of K-means and 10 metrics, were later refined and rigorously tested with additional analyses from multiple aspects, including supervised and unsupervised ML (machine learning) algorithms. The pre-experiment classification with K-means was intentionally made simple and intuitive, and its sole purpose was to formulate the hypothesis intuitively. The rest of the paper was devoted to test the distinctiveness of the four types from multiple aspects.

Once the four classes are obtained the author does further analysis. The evaluated quantities show some unique characteristics for each class proposed in the hypothesis. Additional test further validates the existence of four classes. The results and conclusions drawn are consistent with the data provided in the supplementary material.

The application of supervised machine learning models to identify each VM class seems to perform well and looks consistent with the expectations. More details about the choice of hyperparameter are required.

Response:

Thank you very much for your precise summary! and I fully agree with your characterizations. I have added more detailed parameters based on your suggestions below.

1) When using a neural network what kind of architecture was chosen? What was the values learning rate? Did the author vary the architecture and examined the performance?

Response:

Our network is a simple neural network with the input layer, two fully connected hidden layers and output layer, implemented with TensorFlow framework. The learning rate was not a constant, and the initial value was 0.01. We adjusted learning rate with parameter "patience"=30, which means that if 30 iterations could not improve the model performance, the learning rate would be reduced by 10 times (1/10 of the previous rate).

2) More details about the hyperparameter in various supervised machine learning algorithm is required.

Response:

Besides previously mentioned simple neural network model, we also performed 5 other machine learning (ML) models by using the 'sklearn' package of Python (Version 3.6). K-nearest neighbor model (KNN) requires the number of neighbors, we set it 17. The parameter of SVM is penalty="l2". Random Forest, Gradient Boosting and Logistic Regression don't need hyperparameters when running with sklearn.

3) Have the authors tried performing k-mean with the OTU input data? If yes, do they produce the same four clusters?

Response:

No, I did not do the OTU clustering with K-means, but did so using other machine learning algorithms (*i.e.*, to learn from OTU data directly). The results with the other approach did generate the same four clusters, but with lower precision (Fig 6 & Table S10).

Your question is actually fundamental to the design of my work. The clustering with OTU would be very similar to existing approaches such as Ravel et al. (2012, PNAS) and Gajer et al. (2012: Science Translational Medicine). My conjecture was that we need new metrics, rather than the raw OTUs to classify properly. The raw OTU data certainly contain the raw information, but the "redundancy" in the format of OTU may actually generate "confusing effects" to prevent the dichotomy of healthy vs. disease, or complex vs. simple. With this conjecture, it was unnecessary for me to formulate the initial hypothesis with K-means based on OTUs.

After consequent tests with my 10-metric classification, and it was natural to test whether the tests with other ML methods and OTU would generate similarly good categorization. The test results showed that the performance was not as good as the 10-metric approach, and sometimes are actually bad (around 50% of accuracy, as low as 24%, which actually means either category is possible and unreliable). I would guess that the K-means with OTU would not be good either.

Table S10 in the online supplementary material actually compared the results of both approaches, based on 10-metric vs. OTU-table. Fig 6 also illustrated the comparisons of their precision levels.

Overall the authors need to provide clarification on the above-mentioned choices.

Response:

As a side note, the previous parameters mostly adopted standard default values recommended by the original ML developers. We are refining 2-3 ML approaches (out of the 6 in the paper) for the practical application with more extensive datasets in future. In the future version, we hope to obtain more robust and scalable ML models of possible clinical applications.

Please advise me if you have any further comments and advice about my previous responses!

Reviewer #2 (Comments for the Author):

In this manuscript the author proposes a novel framework for the healthy versus BV vaginal microbiome. Instead of restricting the healthy states to a particular lactobacillus dominance, the author proposes a framework of diversity as a measure of complexity of the VM in both healthy and BV states. The author uses existing VM databases to identify the four proposed states of healthy-simple, healthy-complex, BV-simple, and BV-complex. This hypothesis represents a next step where it considers not simply a dominant species to categorize the VM, but begins to consider the complex interactions among organisms in both the healthy and BV states.

Response:

Please accept my sincere appreciation for your time and efforts on reviewing my work and thank you very much for your precise summary of the manuscript.

Specific comments:

1) Please discuss the decision to limit to 2 states of diversity (simple vs. complex) as clearly the VM is not as "simple" as once thought.

Response:

The decision was mainly based on our previous research experience as well as reading existing literature, and was somewhat intuitive dichotomy of complex vs. simple, further crisscrossing on the BV vs. healthy dichotomy. The next step we used simple K-means clustering algorithm to sketch out the rough categorization of the datasets (into four clusters) and set a foundation to present our hypothesis.

The rest of the paper was aimed to independently test the hypothesis from multiple aspects, including better ML (machine learning) algorithms than the initially used K-

means approach. The previous expert reviewer also echoed a concern on the usage of K-means, and please refer to my answer to that previous question.

In summary, the initial 4-cluster-based hypothesis, although it appears only make sense intuitively, was confirmed by consequent comprehensive independent tests, and no contradictions were found throughout the whole study.

2) When we consider vaginal vaginitis - it is clear that the patient symptoms and laboratory findings are an end state that can be attributed to specific pathogenic organisms. Please address in the discussion the difference in thinking of vaginitis versus vaginosis and why a species specific approach to BV should not be pursued. I agree that it is not likely monotonic, but it is still possible that it is multiple different VM states that constitute BV - this would make a "unifying" hypothesis a no/go from the start.

Response:

I fully agree with your advice. That is, the case of vaginal vaginitis follows simple Koch's postulates—a single pathogen for a specific disease. Initially, BV was postulated to be a polymicrobial disease, but still failed to find a "pathogen cult". Later, BV was characterized as "with dominance by *Lactobacillus*" vs. "lack of dominance", and the presently prevalent hypothesis seems to be the five-type classifications by Ravel et al. (2011, PNAS). In the five-type system, four was considered as associated with healthy and type-IV was deemed of high BV risk. Later studies found that type-IV could be further classified into A & B (Gajer et al. 2012, Science Translational Medicine), and there were also findings that some communities did not fit into any of the existing five types (*e.g.*, Doyle et al. 2018, Applied Environmental Microbiology). I happened to be one of the reviewers of the Doyle et al. (2018), and read the debates of the previous five reviewers who were in split opinions on the results, which were hardly fit into any of the five categories. The experience prompted me to seek alternative hypothesis.

The references mentioned here are cited in my manuscript. Since much of your opinions are also echoed by the previous expert reviewer, I kept my answer here brief. I appreciate very much if you would please also read my previous responses. Please advise me if you have any further comments and advice about my revisions.

June 5, 2023

Prof. Zhanshan (Sam) Ma
Kunming Institute of Zoology Chinese Academy of Sciences
Kunming
Kunming
China

Re: mSystems00049-23R1 (A new hypothesis on BV etiology: dichotomous and crisscrossing categorization of complex vs. simple upon healthy vs. BV vaginal microbiomes)

Dear Prof. Zhanshan (Sam) Ma:

Your manuscript has been accepted, and I am forwarding it to the ASM Journals Department for publication. For your reference, ASM Journals' address is given below. Before it can be scheduled for publication, your manuscript will be checked by the mSystems production staff to make sure that all elements meet the technical requirements for publication. They will contact you if anything needs to be revised before copyediting and production can begin. Otherwise, you will be notified when your proofs are ready to be viewed.

If you would like to submit a potential Featured Image, please email a file and a short legend to msystems@asmusa.org. Please note that we can only consider images that (i) the authors created or own and (ii) have not been previously published. By submitting, you agree that the image can be used under the same terms as the published article. File requirements: square dimensions (4" x 4"), 300 dpi resolution, RGB colorspace, TIF file format.

We recognize that the video files can become quite large, and so to avoid quality loss ASM suggests sending the video file via <https://www.wetransfer.com/>. When you have a final version of the video and the still ready to share, please send it to mSystems staff at msystems@asmusa.org.

Sincerely,

Nicholas Chia
Editor, mSystems

Journals Department
E-mail: mSystems@asmusa.org